# Metabolite changes in blood predict the onset of tuberculosis

January Weiner 3rd[1], Jeroen Maertzdorf[1], Jayne S. Sutherland[2], Fergal J. Duffy [3], Ethan Thompson[3], Sara Suliman[4], Gayle McEwen[1,12], Bonnie Thiel[5], Shreemanta K. Parida[1,13], Joanna Zyla[1], Willem A. Hanekom[4], Robert P. Mohney [6], W. Henry Boom[5], Harriet Mayanja-Kizza [7], Rawleigh Howe[8], Hazel M. Dockrell[9], Tom H.M. Ottenhoff[10], Thomas J. Scriba[4], Daniel E. Zak[3], Gerhard Walzl [11], Stefan H.E. Kaufmann[1] & the GC6-74 consortium[#]

New biomarkers of tuberculosis (TB) risk and disease are critical for the urgently needed control of the ongoing TB pandemic. In a prospective multisite study across Subsaharan Africa, we analyzed metabolic profiles in serum and plasma from HIV-negative, TB-exposed individuals who either progressed to TB 3–24 months post-exposure (progressors) or remained healthy (controls). We generated a trans-African metabolic biosignature for TB, which identifies future progressors both on blinded test samples and in external data sets and shows a performance of 69% sensitivity at 75% specificity in samples within 5 months of diagnosis. These prognostic metabolic signatures are consistent with development of sub-clinical disease prior to manifestation of active TB. Metabolic changes associated with pre-symptomatic disease are observed as early as 12 months prior to TB diagnosis, thus enabling timely interventions to prevent disease progression and transmission.

[1] Max Planck Institute for Infection Biology, 10117, Berlin, Germany. [2] Vaccines & Immunity Theme, Medical Research Council Unit The Gambia at the London School of Hygiene and Tropical Medicine, P. O. Box 273, Banjul, The Gambia. [3] The Center for Infectious Disease Research, Seattle, WA 98145-5005, USA. [4] South African Tuberculosis Vaccine Initiative, Institute of Infectious Disease and Molecular Medicine & Division of Immunology, Department of Pathology, University of Cape Town, Rondebosch 7701, Cape Town, South Africa. [5] Tuberculosis Research Unit, Department of Medicine, Case Western Reserve University School of Medicine and University Hospitals Case Medical Center, Cleveland 44106-4921, OH, USA. [6] Metabolon Inc., Durham, NC 27709, USA. [7] Department of Medicine, School of Medicine, College of Health Sciences, Makerere University, P.O. Box 7072, Kampala, Uganda. [8] Armauer Hansen Research Institute, P.O. Box 1005, Addis Ababa, Ethiopia. [9] Department of Immunology and Infection, Faculty of Infectious and Tropical Diseases, London School of Hygiene & Tropical Medicine, London WC1E 7HT, UK. [10] Department of Infectious Diseases, Leiden University Medical Centre, 2333 ZA Leiden, The Netherlands. [11] NRF-DST Centre of Excellence for Biomedical TB Research and MRC Centre for TB Research, Division of Molecular Biology and Human Genetics, Department of Biomedical Sciences, Faculty of Medicine and Health Sciences, Stellenbosch University, Cape Town, 8000, South Africa. [12]Present address: Leibniz Institute for ZOO and and Wildlife Research, 10315 Berlin, Germany. [13]Present address: Translational Medicine & Global Health Consulting, 10115 Berlin, Germany. These authors contributed equally: January Weiner 3rd, Jeroen Maertzdorf. [#]A full list of consortium members appears at the end of the paper. Correspondence and requests for materials should be addressed to S.H.E.K. (email: kaufmann@mpiib-berlin.mpg.de)

I n 2017, 10 million cases of tuberculosis (TB) disease and 1.6 million deaths due to TB were recorded globally[1], making it the deadliest infectious disease on Earth. A quarter of the world's population is estimated to be latently infected with *Mycobacterium tuberculosis* (*Mtb*), and of these, less than 10% will develop active TB disease during their lifetime[2]. Notably, the risk of TB incidence is 10-fold higher in individuals within the first year after infection[3].

Novel, cost-effective tools for control of TB must include not only new and improved drugs and vaccines, but also assays for rapid and sensitive diagnosis of TB[1]. Defining biomarkers for risk of disease coupled with early and accurate TB diagnosis will enable strategies for prevention and early treatment to prevent progression to advanced disease pathology as well as transmission. Moreover, identifying infected people at high risk of developing TB will facilitate targeted enrollment into drug trials and post-exposure vaccine trials, thus profoundly reducing the number of study participants and trial costs and duration.

Until recently, the only measurable biomarkers associated with increased risk of developing TB were positive TST or IGRA test results[4]. However, these tests have poor specificity for identifying incident TB as over 95% of HIV-negative and ~70% of HIV-positive individuals with TST/IGRA positivity never progress to active TB disease[5]. Mass preventive therapy based on IGRA/TST screening in TB endemic countries would therefore require treatment of 50–80% of the population. This translates to treatment of an estimated 85 people with latent TB to prevent a single case of active TB according to currently available IGRA tests[6], thus putting many healthy individuals at unnecessary risk of adverse events. Such a strategy would neither be cost-effective nor feasible and would not prevent re-infection in high-incidence situations. This was demonstrated by the Thibela trial which enrolled South African mine workers in a setting with an 89% prevalence of latent MTB infection[7] but mass isoniazid preventive therapy did not reduce TB incidence[8].

The Grand Challenges in Global Health GC6-74 project (GC6 project) was initiated in 2003 with the goal of identifying TB biomarkers with prognostic potential. The study encompassed 4462 HIV-negative participants across multiple African field sites (Supplementary Figure 1), reflecting different regions and ethnicities. All participants were household contacts of newly diagnosed TB index cases and were followed for 2 years post-exposure, with blood samples taken at enrollment and at specified follow-up time points. This design provided a unique opportunity to investigate the prospective risk of TB in exposed individuals. The collection of samples from South, West, and East African field sites allowed for comparisons between sites and development of a trans-African biosignature.

Blood transcriptomic biomarkers of TB that discriminate patients from healthy individuals have been identified in several studies[9]. In a recent prospective study[10], a 16-gene transcriptomic signature was identified in the Adolescent Cohort Study (ACS) with the power to predict progression to active TB. The signature was validated with samples from two African sites from the GC6 project showing a sensitivity of 66% and a specificity of 80% in the 12 months preceding the diagnosis of TB. In further pursuance of a transcriptomic risk signature, a combination of 2 gene pairs was found to predict risk of TB at 62% sensitivity and 63% specificity in the tested population[11]. In another promising approach on the same cohort, circulating miRNAs from serum samples were shown to similarly approach 65% specificity at 62% sensitivity[12].

Metabolic profiling has been successfully applied for biomarker discovery in several non-communicable diseases[13–16], but rarely in infectious diseases. To our knowledge, no studies thus far have demonstrated the capability of metabolic profiling in predicting progression to an infectious disease in samples from healthy donors. In TB, metabolic profiling was found to discriminate between TB patients and healthy individuals[17,18], and our previous study identified a metabolomic biosignature which discriminates patients from healthy controls with remarkably high accuracy[19] (AUC > 0.98; 95% CI: 0.97–1.00).

Here, we investigated longitudinal changes in metabolic profiles in serum and plasma from household contacts of adults with pulmonary TB who either remained healthy (controls) or developed TB (progressors) and applied machine learning techniques to discover metabolite signatures that predict risk of progression to TB across Africa. Amongst recruited individuals, 2.2% progressed to TB (progressors) whilst the rest remained asymptomatic until the end of the 2 years observation period (controls). All analyzed blood samples from household contacts were collected before TB diagnosis and therefore represent clinically asymptomatic individuals.

Two hypotheses were tested: (i) are there metabolites that can predict progression from infection to TB; and, if yes, (ii) does prediction rely on innate metabolic risk factors, or on metabolic processes occurring during disease progression? Accordingly, predictive metabolites fell into the following classes: (a) metabolites that reflect baseline (BL) risk factors and show a consistently significant difference between progressors and controls. We term these risk-associated metabolites, as these indicate a higher likelihood of progression to TB; (b) metabolites predictive of active TB, which show time-dependent differences between study groups, indicating progression to disease. We term these disease-associated metabolites, as the absolute difference in abundances between progressors and controls increases towards clinical manifestation of TB implying that these metabolites are indicators of the host response to subclinical TB.

## Results

**Study cohorts**. To aid the detection of biomarkers which would be potentially applicable across the African continent, study participants were recruited at field sites in East, West, and South Africa (Fig. 1).

Within the GC6-74 cohorts, 4462 HIV-negative healthy household contacts of 1098 index TB cases were recruited from 2006 to 2010 with the follow-up completed in 2012 at four African sites included in this study (Fig. 1), i.e., SUN (Stellenbosch University, South Africa), MRC (Medical Research Council Unit, The Gambia), AHRI (Armauer Hansen Research Institute, Ethiopia), and MAK (Makerere University, Uganda).

A total of 97 individuals who developed active TB within the 2 year follow-up period (progressors) were included in this study and matched at a ratio of 1:4 with participants who remained healthy during the 2-year follow-up period (controls). A total of 751 serum or plasma samples from these individuals were analyzed using untargetted mass spectrometry (Supplementary Table 5).

Initial samples collected upon enrollment were termed baseline (BL) samples. Further samples were taken 6 and 18 months post-exposure, provided that the participant had remained TB free at the time of sample collection. The progressor samples were also retrospectively labeled for time to TB, i.e., the number of months prior to actual diagnosis of active TB (as opposed to time post exposure). Before the analysis, for each site, samples from two thirds of all individuals were selected as training set, and the remaining samples as a blinded test set.

**Biosignature model building and validation**. We pursued an a priori determined analysis strategy. The design and validation of biosignatures derived from metabolic profiling comprised three

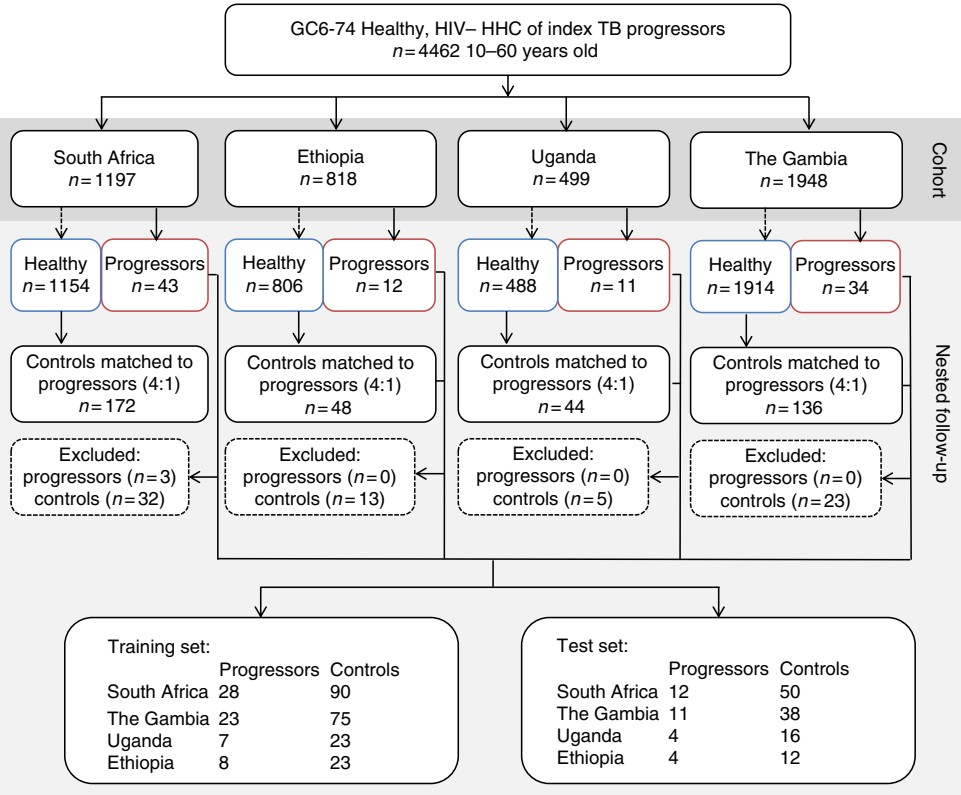

**Fig. 1** Consort diagram for the study. The samples were collected at: SUN, Stellenbosch University, South Africa; MRC, Medical Research Council Unit, The Gambia; AHRI, Armauer Hansen Research Institute, Ethiopia; MAK, Makerere University, Uganda

stages: (i) generate signatures (machine learning models) based on training set samples only; (ii) validate the models using blinded samples from the test set; (iii) further validate the findings using external, independent data sets.

In the first step, we used the training set samples to optimize the machine learning procedure. We used 10-fold cross-validation as a measure for internal evaluation of the models on the training set (Supplementary Figure 3, Supplementary Table 6), and we tested to what extent the machine models in the training set were predictive between sites (Supplementary Figure 4, Supplementary Table 7).

Finally, once we had ensured that the methodology produced significant predictions in a cross-validation test within the training set and that there was a comparability between results from different sites and sample types, we trained random forest models[20] comprising the entire training set (model Total) or BL samples only (model Total/Baseline).

Each biosignature was then validated by making a blinded prediction on the test set only. All models and results are summarized in Supplementary Tables 10 and 11.

Figure 2a and b shows the performance of the universal models (Total and Total/Baseline, respectively) on the final validation set. The Total model significantly validates on the overall validation set, including all four sites, for both proximate samples (< 5 months to TB diagnosis; AUC: 0.78; 95% CI: 0.62–0.94, Wilcoxon $q = 0.0033$), and distal samples (≥ 5 months to TB diagnosis; AUC: 0.68; 95% CI: 0.58–0.79; Wilcoxon $q = 0.0033$; see Supplementary Table 10). Assuming a required minimum of 75% specificity[21], this corresponds to 53% sensitivity for all samples, and 69% for proximate samples.

The signatures also validated on the South Africa and The Gambia cohorts independently (Supplementary Table 11). While these signatures did not significantly validate separately on

samples from the smaller Ethiopia and Uganda cohorts, which contained only four TB-progressors in each test set, the Total/Baseline signature did validate on proximate samples if these two cohorts were considered jointly (AUC: 0.68; 95% CI: 0.51–0.85).

The high performance on the proximate samples was not due to samples collected within days from the diagnosis. If samples collected 1 month or less before the diagnosis time point were excluded from the analysis, the model performance in the proximate data set increased to an AUC of 0.82 (95% CI: 0.57–1.00).

We next scrutinized the constructed models to understand what classes of metabolites were discriminatory. To this end, we applied an enrichment test to metabolites ordered by their relative importance in either of the models Total and Total/Baseline (Supplementary Table 9). We tested the enrichment of 42 sets of modules which included both, categories of biochemical compounds (such as amino acids) as well as clusters of metabolites identified in TB in prior work[19]. We found significant enrichment of amino acids (CERNO test $q < 0.01$ for both models) as well a significant enrichment in the cluster containing glycocholate, taurocholenate, kynurenine, and cortisol (CERNO test $q < 0.05$; Fig. 2c) in the Total/Baseline model. Among the top 25 compounds, the disease-associated markers dominated (Supplementary Table 9).

**Validation with independent data sets**. To further test our findings, we sought validation in independent data sets. Having identified a metabolic signature of risk for TB which increased during TB progression, we asked whether there is a common biological denominator between progression toward TB and clinically apparent TB. We hypothesized that in such a case, the changes of metabolites that we have previously described[19] for TB

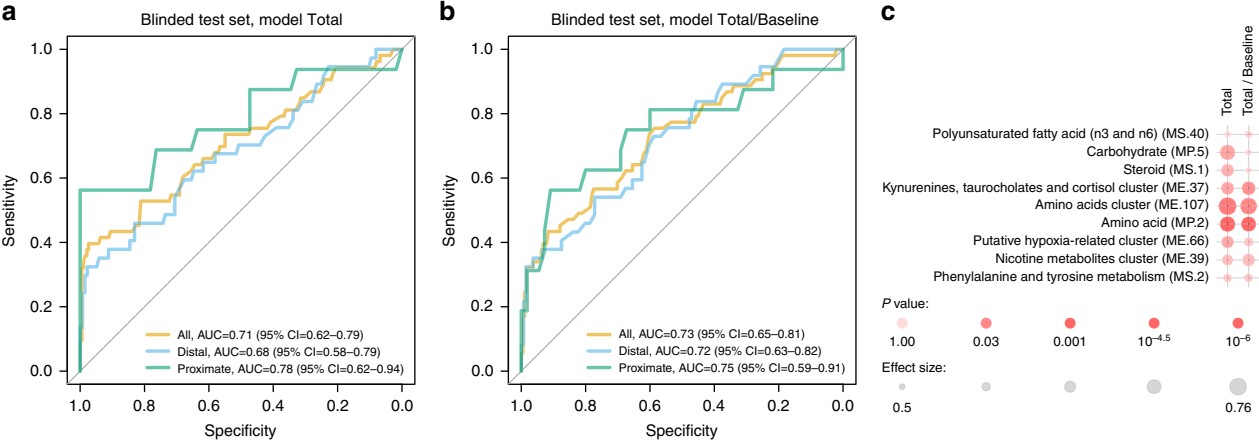

**Fig. 2** Machine learning models (biosignatures) discriminating between progressors and controls. Panels show receiver–operator characteristic (ROC) curves. The three panels correspond to the three models tested: **a**, model Total which was generated using all training set samples; **b**, model Total/Baseline which was generated using only BL training set samples. Model evaluation was stratified by time to TB diagnosis: all, evaluation on all test set samples; proximate, evaluation on test set samples collected < 5 months before TB diagnosis; distal: ≥ 5 months. **c** Results of enrichment test on metabolites ordered by their importance in the Total and Total/Baseline models. The metabolite sets correspond to biochemical groups and clusters of metabolites identified previously in TB patients. Color intensity corresponds to p-value, and symbol size corresponds to the strength of the enrichment. P-values were corrected for multiple testing, and AUC was used as a measure of effect size

could also predict risk of progression from infection to disease. To test this, we used the data set from healthy individuals and tuberculosis patients described earlier[19]. This data set is referred hereafter as TB-HEALTHY data set.

We first asked whether the models derived from all individuals in the GC6-74 training set (model Total) can also correctly classify patients suffering from TB even though all individuals in the GC6 training set were asymptomatic. To this end, we have applied the Total model described above to the TB-HEALTHY data set. Indeed, our Total model discriminated healthy individuals from active TB cases (AUC 0.92; 95% CI: 0.87–0.97; Supplementary Figure 5).

Furthermore, we asked the reverse question, whether metabolite profiles of TB patient samples can be used to predict progression toward disease in healthy individuals. Here, we tested a model derived from TB-HEALTHY on the prospective GC6-74 samples (both training and test samples, since the TB-HEALTHY data set is completely independent). Serum metabolite levels from 44 TB patients and 92 controls from the TB-HEALTHY data set were used to train a random forest classifier, which was then directly applied to the longitudinal data from the present study.

The TB-HEALTHY signature was significantly predictive for the GC6-74 data (overall AUC 0.68; 95% CI: 0.64–0.73, corresponding to 50% sensitivity at 75% specificity) and similarly showed a stronger performance for proximate samples (overall AUC 0.82; 95% CI: 0.75–0.89, 73% sensitivity at 75% specificity; Fig. 3a). For the largest sample sets, from The Gambia and South Africa, the performance on proximate samples showed AUCs of 0.86 (95% CI: 0.75–0.96) and 0.81 (95% CI: 0.69–0.93), respectively (Supplementary Table 12). Intriguingly, the TB-HEALTHY model performed at least as well as the biosignatures derived from the GC6-74 study itself.

Both TB-HEALTHY and Total models included a number of shared metabolites as strongest predictors. Among the top twenty predictors from both models, sixteen were shared between the two models, including kynurenine, cortisol, bile acids, 3-carboxy-4-methyl-5-propyl-2-furanpropanoate acid (CMPF), and tryptophan, and variable importance score (mean decrease in Gini coefficient) was significantly correlated between both models (Pearson correlation, 0.89, $p = 0.00$). These results support the

hypothesis that the prognostic biosignature of subclinical TB is similar to the signature for diagnosis of TB.

**A model with 10 features predicts TB progression**. The models created hitherto were based on all available features present in the data set; however, for a practical implementation, a much lower number of variables is required. On the other hand, reducing the number of features negatively impacts the performance of a model.

We have determined the relationship between the number of features used in a model for the TB-HEALTHY and the model performance data set using leave-one-out cross-validation (Supplementary Figure 6) and, based on this, we have generated post-hoc a model reduced to only 10 features, including five disease-associated metabolites and one risk-associated metabolite (unidentified features were excluded from the model). While the reduction of the number of features decreased the observed performance of the reduced model on the validation set, it was similar to the model including all features and had significant predictive power for all cohorts except the Uganda samples (Fig. 4 and Supplementary Table 13).

**Metabolomic signatures are specific for TB**. We next determined whether the predictive signatures identified in the GC6-74 study specifically discriminates TB from other respiratory diseases (ORD), since observed changes of metabolites such as cortisol could reflect a more general inflammatory state rather than a specific TB signature. To test this hypothesis, we collected an additional independent set of plasma samples from The Gambia from patients reporting with symptoms suggestive of active TB. These patients were later on diagnosed with either TB or ORD (including chronic obstructive pulmonary disease (COPD), asthma, pneumonia, and other respiratory tract infections).

We applied the Total model trained on all samples from the GC6-74 study to the metabolic profiles from TB and ORD patients within this separate Gambian cohort. We observed a specific and sensitive discrimination between TB and ORD (AUC: 0.87; 95% CI: 0.80–0.93; Supplementary Figure 5),

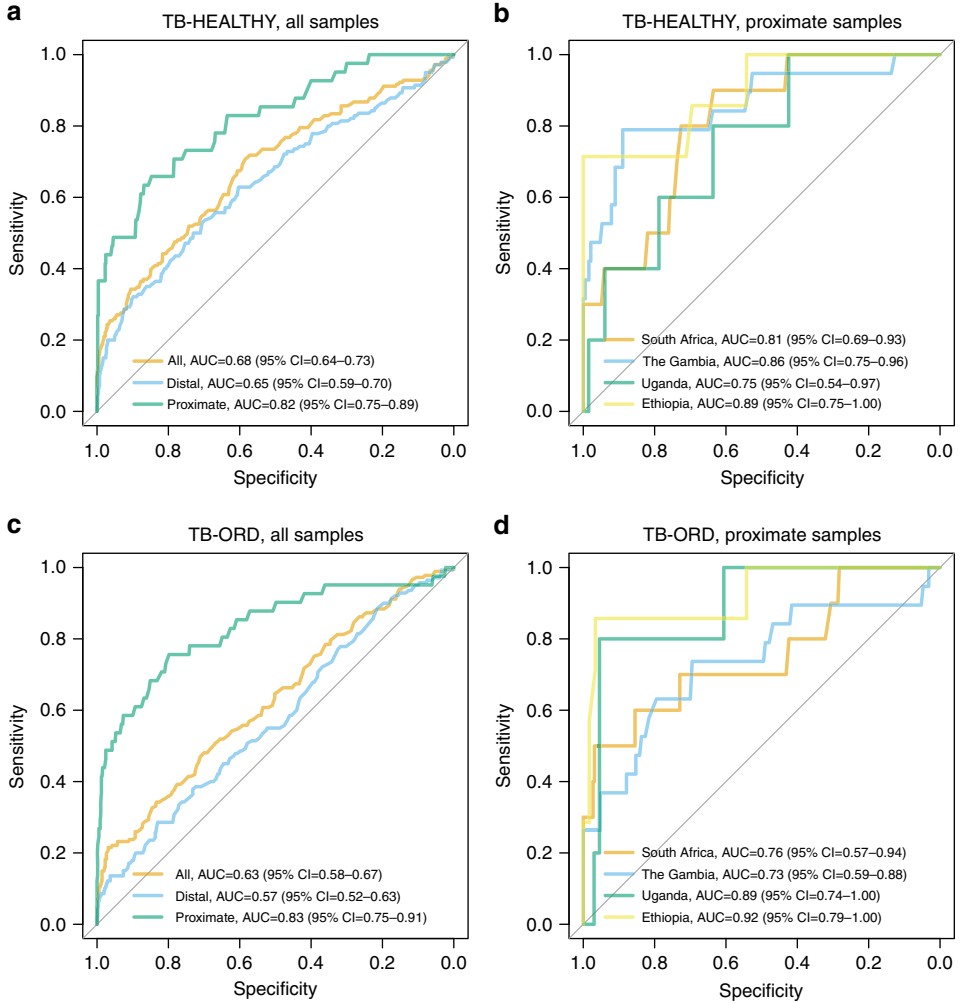

**Fig. 3** Predictive power of external metabolomic signature applied to the full GC6-74 data set. Panels **a** and **b** show the TB-HEALTHY model derived from the sera of TB patients and healthy individuals, while panels **c** and **d** show the TB-ORD (TB vs. other respiratory diseases) model derived from plasma samples of TB patients and from plasma samples of patients suffering from other respiratory diseases. Panels **a** and **c** show models applied either to all samples (all), or samples stratified by time to diagnosis (proximate and distal). Panels **b** and **d** show the results stratified by site

indicating that the predictive models detect disease-specific biology.

Again, we reversed this procedure, testing whether metabolic profiles derived from the TB-ORD data set can be used to detect the progression to TB in healthy individuals. We constructed a random forest machine learning model based on TB-ORD samples and applied it to classify the GC6-74 samples. The model showed substantial predictive power (Fig. 3c, d) with AUC for proximate samples ranging from 0.73 to 0.92 (see Supplementary Table 14) with six predictors shared with the Total model. The variable importance of the metabolites was significantly correlated between the TB-ORD and Total models (Pearson correlation, 0.69, $p = 0.00$). This demonstrates that the same metabolites that specifically distinguish TB patients from ORD patients—including cortisol and CMPF—also distinguish progressors from controls.

We conclude that the signature differentiating TB progressors from controls represents an alteration in metabolic state specific to TB pathology.

**Temporal changes and time-independent profiles**. To better understand metabolic changes and biological mechanisms

underlying TB progression, we used linear modeling to identify individual metabolites (i) that significantly differ in relative abundance between progressors and controls and (ii) that show a significant increase over time in progressors only.

Observed differences between progressors and controls were consistent with previously published differences between active TB and healthy or latent TB-infected individuals[19], including alterations in the relative abundances of particular amino acids, bile acids, and cortisol. For example, several amino acids, including histidine (generalized linear model, GLM, $q = 7.7 \times 10^{-05}$), alanine (GLM $q = 7.7 \times 10^{-05}$), and tryptophan (GLM $q = 0.00022$) had significantly lower abundances in the progressor group, while cortisol (GLM $q = 1.7 \times 10^{-05}$) was higher in progressors.

Clear differences between progressors and controls were more prominent in the proximal than in the distal samples (Supplementary Table 15, Fig. 5, Supplementary Figure 7). This was confirmed by significant dependence of metabolite abundances from time to diagnosis (Supplementary Table 16). Figure 5 illustrates the significant differences in temporal regulation of metabolites. Cortisol and kynurenine levels in progressors begin to deviate from controls at about 12 months prior to active TB. In contrast, the amino acids

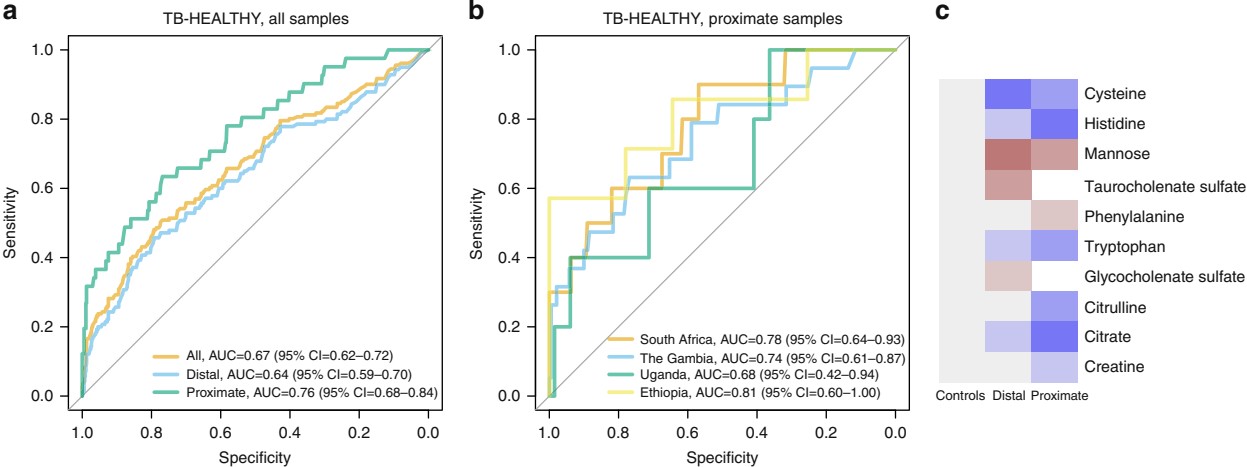

**Fig. 4** The trans-African signature based on TB-HEALTHY data set reduced to 10 metabolites. **a** performance of the model in the total, distal and proximate samples; **b** performance of the model for proximate samples at the four African sites; **c** list of metabolites included in the model and their relative abundance compared to controls. Colors correspond to scaled average abundances relative to the average in controls

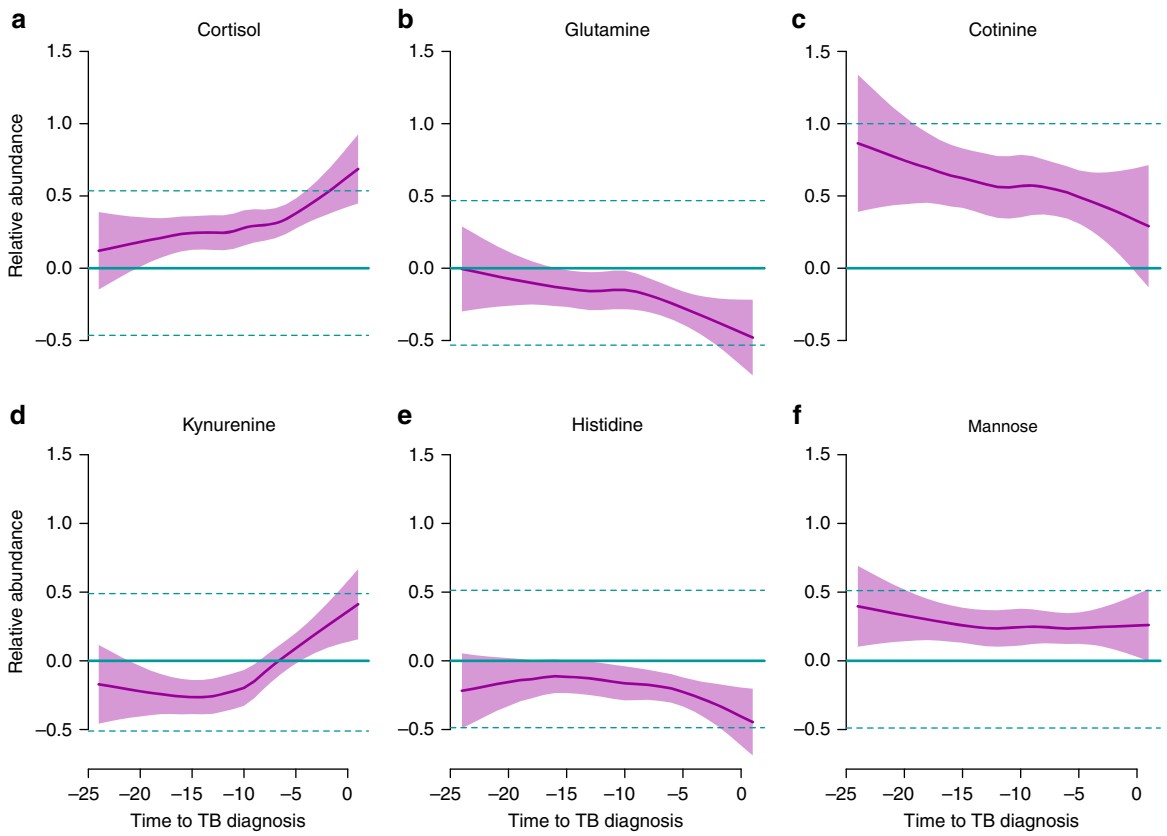

**Fig. 5** Profiles of four selected metabolites revealing changes in abundances in progressors. **a**, **b**, **d**, **e**: disease-associated metabolites; **c**, **f**: risk-associated metabolites. Shaded area indicates 95% confidence intervals. Solid green line indicates median for controls and dashed green lines indicate first and third quartiles for controls

histidine and glutamine start to deviate by 9 and 6 months prior to clinical TB (Fig. 5b, e).

Kynurenine is a crucial metabolite of the indoleamine 2,3-dioxygenase pathway thought to play a critical role in immunoregulation of TB, and was among the most prominent markers for active TB in our previous study. Although we did not find any significant differences in the relative abundance of kynurenine when the progressors were compared to controls, the increase of kynurenine over time to TB diagnosis was significant (linear model $q = 2 \times 10^{-05}$).

Enrichment testing on compounds significantly increasing over time to TB diagnosis showed a significant enrichment for long chain fatty acids (CERNO test $q = 5.6 \times 10^{-05}$, AUC = 0.77), cluster of kynurenines, taurocholates, and cortisol (CERNO test

$q = 5.6 \times 10^{-05}$, AUC = 0.81), and lipids (CERNO test $q = 0.00095$, AUC = 0.64; Supplementary Table 17).

Finally, we attempted to identify risk-associated metabolites. To this end, we searched for metabolites with consistent, time-independent differences between progressors and controls at time points distal from TB diagnosis. Two metabolites were differentially abundant at times far from diagnosis of active TB (Supplementary Figure 7). Cotinine, a xenobiotic metabolite of nicotine was consistently more abundant in progressors than in controls even more than a year before the diagnosis (Fig. 5c; GLM $q = 0.0064$). The presence of cotinine correlated with both, smoking status and smoking intensity (see Supplementary Figure 8), illustrating the ability of metabolic profiling to identify static environmental risk factors simultaneously with dynamic disease processes. Indeed, smoking alone was a predictor for progression (OR = 2, $\chi^2$ $p = 0.00042$), especially in the SUN cohort, where smoking was common (OR = 4.6, $\chi^2$ $p = 3.1 \times 10^{-07}$). In addition, progressors showed an increased abundance of mannose in samples collected immediately post-exposure (linear model $q < 10^{-4}$).

## Discussion

The ability to detect TB at an early stage after exposure to *Mtb*, but before clinical symptoms arise, allows early intervention needed for control of the continuing pandemic. Biosignatures that indicate risk factors or signs of "preclinical TB" in otherwise healthy individuals could be harnessed for early treatment to prevent clinical disease and dissemination. Here, we demonstrate that changes in serum or plasma abundances of small metabolic compounds identify individuals who progressed to clinical TB. The magnitudes of these changes increased as disease onset approached. Several metabolites associated with TB progression in this study had been found previously to differ between TB patients and healthy individuals in a previous study[19]. By comparing TB patients with patients suffering from other pulmonary diseases, we here demonstrate that these metabolite differences are specific to TB. Accordingly, the identified metabolomic signatures demonstrate specific and robust performance in predicting subclinical TB and progression to active TB.

We further demonstrate the utility of metabolic profiling for predicting progression to TB by successful application of a TB diagnostic signature derived from an independent study cohort[19]. Both the descriptive analysis of changes of serum abundances and the comparison of the machine learning models show that changes in concentrations of metabolites in progressors were well aligned to differences in these metabolites between TB patients and healthy individuals. This strongly supports the hypothesis that metabolic profiling identifies subclinical TB. In fact, for proximate samples, at 75% specificity, the sensitivity of these models approached (TB-HEALTHY, 73%) or exceeded (TB-ORD, 76%) the proposed requirements for a target product profile for the development of a test for predicting progression to active TB disease[21].

Interestingly, the performance of models based on external data sets was better than that of the models derived from the progressors vs. controls of the GC6 cohort. This can be explained as follows: the signature derived from asymptomatic individuals is based on a less pronounced phenotype of a slowly emerging TB, which might lead to a noisy signature. In contrast, a clearly defined signature based on TB patients who all show the molecular markers of symptomatic, clinical TB performs well when applied to noisy data. A similar phenomenon has been observed for transcriptomic data[22], where the more pronounced TB signatures from HIV-positive patients performed better even when applied to the noisier data from HIV-negative patients.

Despite differences in sample type, life style, genotype, diet, etc. between the different cohorts, a single predictive metabolic signature predicted progression across these cohorts and populations. Consistent with this, the TB-HEALTHY signature derived from a cohort in South Africa showed a strong performance when applied to proximate samples from The Gambia (AUC: 0.86; 95% CI: 0.75–0.96) and Ethiopia (AUC: 0.89; 95% CI: 0.75–1.00).

In this study, we have not included HIV-positive individuals by design, even though the interplay between these two diseases plays a pivotal role in TB epidemiology. It is unknown to what extent the presence of opportunistic infections and the perturbed immune responses associated with HIV infection will alter the predictive signatures. However, previous studies show hardly any overlap between the TB and HIV metabolic profiles[23,24], with plasma glutamate being the only biomarker common for both TB and HIV[24].

While both tuberculin skin test positive (TST[+]) and negative (TST[−]) individuals were enrolled in the study, we observed less than 10 conversion events in the whole study and could not find any conclusive evidence for a link between TST conversion and metabolite profiling (see "Methods" for details).

Temporal changes in metabolite levels between progressors and healthy controls were concordant with the hypothesis that metabolic profiling detects subclinical disease in progressors, rather than capturing a set of stable risk-associated markers. For example, the abundances of amino acids that were previously shown to be decreased in TB[19] showed a gradual decline in progressor samples approaching clinical diagnosis (Fig. 5). This has been further corroborated by comparing the signatures of progressors with the signatures of TB patients, suggesting a quantitative rather than qualitative change from the subclinical stage of TB development preceding clinical diagnosis to clinical TB, at least at the molecular level.

In contrast to metabolites characteristic for TB, cotinine and mannose were detected at elevated levels irrespective of time to TB onset. As a nicotine metabolite, cotinine is associated with smoking, a well defined risk factor[25] for the development of active TB and can act synergistically on disease risk with alcohol consumption[26]. The higher basal levels of cotinine and related metabolites corresponds to the fact that reported smoking was higher in the progressors from the South African cohort than in the predominantly Muslim population of The Gambia. Interestingly, the levels of cotinine approached the levels in the healthy population at time of TB diagnosis, which tempts us to speculate that smoking intensity decreases with disease progression. While cotinine as a biomarker is redundant (as it merely reflects smoking status), it does serve as a positive control, demonstrating that risk factors can be identified with our approach.

Mannose, another metabolite showing differences between progressors and controls irrespective of time to diagnosis, plays a central role in mammalian energy generation and regulation, and can have both beneficial and detrimental effects[27]. The consistently elevated levels of mannose in progressors may hint to an impaired glucose tolerance or insulin resistance[28,29] and could even be associated with an inherent risk of developing type 2 diabetes[30] in these individuals. The observation of differences in mannose and cotinine levels at basal time points emphasizes that metabolic markers of risk could be detected by metabolic profiling.

C-glycosyltryptophan has previously been observed to be negatively correlated with lung function[31]. The progressors in our study showed significant differences in the abundance of this compound during the time prior to diagnosis. This might reflect impaired lung function as a result of inflammatory responses during disease progression.

Decreasing glutamine levels are observed under inflammatory conditions and this nonessential amino acid may become essential during infection and disease[32,33], such that dietary supplementation of glutamine can be beneficial in some patient populations[34]. Glutamine is required for the proper functioning of the immune system and during mycobacterial infection lymphocytes, neutrophils, and macrophages rapidly consume glutamine[32,35]. In this respect, the gradual drop in glutamine levels observed in progressors likely reflects increasingly exacerbated lung pathology in these individuals.

Changes in amino acid and cortisol levels can be detected as early as 12 months before disease onset, becoming even more prominent toward clinical diagnosis of TB. We conclude that manifestation of active TB is the apex of a prolonged process which remains subclinical for many months. Since these metabolomic changes can already be detected during the asymptomatic phase, metabolic profiling allows stratification of TB risk in individuals with latent TB into high- and low-risk individuals, as was recently shown for blood transcriptomic signatures of TB risk[10–12]. The metabolomic signatures identified here can potentially be combined with transcriptomic signatures to further improve sensitivity and specificity of TB risk prediction[11]. This signature can identify high-risk individuals in the absence of available sputum for microbiological diagnosis, facilitating treatment prior to development of disease pathology when the bacterial load and the likelihood of disease transmission is low. A proof of concept trial is currently underway stratifying participants based on the 16-gene transcriptomic correlate of risk[10] to test its potential for targeted intervention (clinicaltrials.gov identifier: NCT02735590). While the development of a diagnostic test for a metabolite can be a costly process, such tests are already available for a number of relevant compounds such as cortisol.

Along with identifying high-risk individuals for prophylactic treatment, these risk signatures have potential value for clinical trials of new intervention measures. Selecting such individuals for participation has the potential to increase the power and benefit of clinical trials, reducing participant numbers, and trial duration, thereby lowering trial cost and increasing trial effectiveness. Furthermore, the biological insights provided by metabolic profiling in addition to peripherial blood transcriptomics may aid in the development of host-directed therapies.

Undoubtedly, before practical point-of-care application of a metabolomic signature further studies are needed. An important consideration for metabolomic studies is the availability of inexpensive quantitative procedures for the metabolites of interest[36]. As some of these are readily available (e.g., cortisol), a follow-up study should focus on those metabolites which can be determined by simple procedures, using our data set as a guide line.

Furthermore, our study did not include samples from the progressors collected after clinical diagnosis and instead relied on separate data sets which included TB patients. Hence, a detailed time course characterization of patients before clinical diagnosis as well as during and after treatment would be an important step to corroborate our results and to progress toward practical implementation of a metabolic signature.

Blood metabolomic profiles are not exclusively dependent on processes ongoing in peripheral blood cells, but can also provide biological information on host–pathogen interplay at the site of disease and in other tissues[37]. We are confident that a trans-African prognostic signature for TB consisting of disease-associated and specific metabolites can be constructed. Thus, our metabolomic signature will contribute both to TB control and to better understanding of TB pathogenesis.

## Methods

**Study design and participants**. We recruited 4462 HIV-negative healthy household contacts of 1098 index TB cases across in the GC6-74 cohorts in four African sites included in this study. We enrolled 1197 contacts of 209 index cases in South Africa (SUN) between February 27th, 2006 and December 14th, 2010, 1948 contacts of 402 index cases in The Gambia (MRC) between March 5th, 2007 and October 21st, 2010, 818 contacts of 154 index cases in Ethiopia (AHRI) between February 12th, 2007, and August 3rd, 2011, and 499 contacts of 181 index cases in Uganda (MAK), between June 1st, 2006 and June 8th, 2010. Follow-up visits in the GC6-74 household contacts cohorts concluded on November 28th, 2012 in South Africa, October 22nd, 2012 in The Gambia, August 16th, 2012 in Ethiopia, and May 4th, 2012 in Uganda.

The study includes several cohorts with varying study designs and geographic sites, all with a prospective longitudinal design to identify prospective correlates of risk of TB. All sites adhered to the Declaration of Helsinki and Good Clinical Practice guidelines in the treatment of all study participants.

The household contact study design included participants from four African sites: South Africa, The Gambia, Ethiopia, and Uganda, as part of the Bill and Melinda Gates Grand Challenges 6–74 study. The GC6-74 cohorts consisted of 4462 HIV-negative participants, aged 10–60 years, with no clinical signs of pulmonary TB. Participants had to be household contacts of an index TB case, who was at least 15-year-old, with a confirmed positive sputum smear for acid fast bacilli, diagnosed within the last 2 months. For all sites, adult participants, or legal guardians of participants aged 10–17 years old, provided written or thumb-printed informed consent to participate after careful explanation of study aims and any potential risks.

Participants who progressed to active TB disease within the 2-year follow-up period were considered progressors (TB classifications A-K, Supplementary Tables 1, 2). For the TB-ORD validation set, the individuals were classified as TB if they were culture positive using Mycobacteria Growth Indicator Tube, or as ORD, if they were confirmed culture-negative. Of 145 patients, 124 (86%) had an undefined respiratory tract infection; 6 (4%) had bacterial pneumonia, 4 (2.8%) had COPD; 2 (1.4%) had asthma; 1 (0.7%) had emphysema, and 4 (2.8%) had no final diagnoses. All were followed for 2 months and checked for clinical improvement. In addition, all subjects were confirmed culture-negative (40 days of culture) with two separate samples to exclude the possibility of TB.

Study exclusion criteria were current or previous anti-retroviral treatment, history of TB, pregnancy, participation in drug and/or vaccine clinical trials and chronic disease diagnosis or immunosuppressive therapy within the past 6 months, and living in the study area for less than 3 months. Furthermore, participants who developed incident TB were only included in the study if they developed incident TB disease 3 months after enrollment. This was to ensure that no-one had undiagnosed clinical TB at the time of household contact and collection of the baseline sample. If a person had TB at any point in their lifetime before the GC6-74 study, they were excluded from enrollment into the study. A positive HIV rapid test was furthermore an exclusion criterion of samples from this study. The percentages of individuals who completed month 24 examination were 87% for SUN, 84% for MRC, and 80% for MAK.

Each progressor was matched to four non-progressors/controls, who remained healthy during follow-up, by site, age class, sex, and wherever possible year of recruitment (classifications R and S, Supplementary Tables 1, 2). Age included four classes: <18, 18–25, 25–36, and >36 years of age, and year of enrollment had three categories: 2006/2007, 2008, and 2009/2010.

The South African cohort was recruited from the communities of Ravensmead, Uitsig, Adriaanse, and Elsiesriver and clinical sites affiliated with the University of Stellenbosch and Tygerberg Hospital Infectious Disease Clinic in Cape Town, South Africa. The study protocol was approved by the Stellenbosch University Institutional Review Board (ref no. N05/11/187). South African participants do not receive isoniazid preventative treatment per South African national treatment guidelines. Samples were collected from participants at enrollment (baseline samples), and 18 months. The Gambian cohort was recruited from the Greater Banjul area and Medical Research Council (MRC) outpatient departments in The Gambia. The site protocol was approved by the Joint Medical Research Council and The Gambian Government ethics review committee, Banjul, The Gambia; (reference no. SCC.1141vs2). The Ethiopian cohort was recruited from Arada, T/ Haimanot, Kirkos, and W-23 clinical centers in Addis Ababa, Ethiopia. The site protocol was approved by the Armauer Hansen Research Institute (AHRI)/All Africa Leprosy, TB, and Rehabilitation Training Center (ALERT) ethics committees; reference no. P015/10. Finally, the Ugandan cohort was recruited from the Uganda National Tuberculosis and Leprosy Program treatment center at the Old Mulago Hospital and surrounding communities in Kampala, Uganda. The site protocol was approved by the ethics committees of University Hospitals Case Medical Centre (reference no. 12-95-08) and the Uganda National Council for Science and Technology; (reference no. MV 715); these participants received preventative treatment. For these three sites, samples were collected at enrollment (baseline), 6 and 18 months post-enrollment. Samples from all four sites were shipped to the central biobank at the University of Cape Town for analysis, and processing was approved under the University of Cape Town Human Research Ethics Committee HREC; reference no. 013/2013 (Supplementary Table 3).

All samples were collected from individuals who were asymptomatic at the time of their clinical exam.

**Tuberculin skin test (TST)**. In a majority of individuals, a tuberculin skin test (TST) was conducted at baseline. In the South African and Ugandan subsets, the vast majority of the individuals showed a positive response (TST$^+$, ≥ 10 mm) already at the baseline (Supplementary Table 4), and more individuals converted throughout the study. In the Ugandan and Gambian subsets, ≥ 40% individuals were TST$^+$. For eight individuals (six from The Gambia and two from South Africa), samples before and after conversion were available.

We have tested post hoc for association between TST and metabolite abundances. First, we used paired Wilcoxon test to compare samples before and after conversion for the control individuals for which before and after conversion samples were available. Then, we tested Spearman correlation between the TST size reported and the abundances of compounds at baseline. Next, we compared the abundances of compounds in control individuals with TST ≤ or ≥ than 10 mm by Wilcoxon test. In both tests, the p-values were corrected for multiple testing using the Benjamini–Hochberg method. Finally, we trained random forest machine learning models on the compound data for discrimination between TST$^+$ and TST$^-$.

Wilcoxon or Spearman tests revealed no differences in any of the comparisons. For the two main sets (South Africa and The Gambia), as well as for one of the small sets (Uganda), we found no evidence of any link between TST results and metabolic profiles. Random forest model cross-validated on the 32 baseline control samples from Ethiopia, however, revealed a significant discrimination between the TST$^+$ and TST$^-$ individuals (AUC: 0.90; 95% CI: 0.78–1.00, q-value $4.7 \times 10^{-05}$) even though individual compounds did not significantly differ between TST$^+$ and TST$^-$ individuals. The model did not validate when applied to the other data sets (South Africa, The Gambia, Uganda) or the remaining samples from Ethiopia.

**Training and test set**. Prior to analysis, progressor samples were divided between test and training sets in such manner that the resulting sets had identical stratification in respect to age, sex, and sample time to TB (Supplementary Figure 2). For each sample, at most four (where available) matched control samples from different donors were selected. These sets were locked and the test set was blinded. All analyses, including metabolomic profiling and bioinformatic analyses, were performed first on the blinded dataset. Machine learning models derived from the training set were applied to the test set and locked prior to unblinding.

**Metabolic profiling**. Plasma was derived from ficoll separation of blood samples during PBMC isolation. Serum was derived from clotted blood tubes. For serum collection, SST Vacutainer tubes from BD were used, centrifuged for 10 min at 2500×g within 2 h of blood draw, aliquoted, and stored at −70 °C until analysis. Ugandan plasma samples were diluted in RPMI. Samples were stored at –80 °C until processed. TB-ORD plasma samples were derived from heparinized blood following centrifugation and frozen at −20 °C prior to shipment.

Sample preparation was carried out at Metabolon, Inc. as follows[38]: recovery standards were added prior to the first step in the extraction process for quality control purposes. To remove protein, dissociate small molecules bound to protein or trapped in the precipitated protein matrix, and to recover chemically diverse metabolites, proteins were precipitated with methanol under vigorous shaking for 2 min (Glen Mills Genogrinder 2000) followed by centrifugation. The resulting extract was divided into four fractions: one for analysis by reverse phase ultra-performance liquid chromatography–tandem mass spectrometry (UPLC-MS/MS; positive ionization), one for analysis by reverse phase UPLC-MS/MS (negative ionization), one for analysis by gas chromatography–mass spectrometry (GC-MS), and one sample was reserved for backup. For the TB-ORD samples (i.e., TB vs. other respiratory diseases), the resulting extract was divided into five fractions: two for analysis by two separate reverse phase UPLC-MS/MS methods with positive ion mode electrospray ionization (ESI; "UPLC-MS/MS Pos Early" and "UPLC-MS/MS Pos Late"), one for analysis by reverse phase UPLC-MS/MS with negative ion mode ESI ("UPLC-MS/MS Neg"), one for analysis by Hydrophilic Interaction Liquid Chromatography (HILIC)/UPLC-MS/MS with negative ion mode ESI ("UPLC-MS/MS Polar"), and one sample was reserved for backup.

Three types of controls were analyzed in concert with the experimental samples: samples generated from a pool of human plasma extensively characterized by Metabolon, Inc. served as technical replicate throughout the dataset; extracted water samples served as process blanks; and a cocktail of standards spiked into every analyzed sample allowed instrument performance monitoring. Instrument variability was determined by calculating the median relative standard deviation (RSD) for the standards that were added to each sample prior to injection into the mass spectrometers (median RSD = 3–5%; n ≥ 30 standards). Overall process variability was determined by calculating the median RSD for all endogenous metabolites (i.e., non-instrument standards) present in 100% of the pooled human plasma samples (median RSD = 9–12%; n = several hundred metabolites, depending on the matrix tested). Experimental samples and controls were randomized across the platform run.

**Mass spectrometry analysis**. For non-targeted MS analysis, extracts were subjected to either UPLC-MS/MS or GC-MS. The chromatography was standardized and, once the method was validated, no further changes were made. As part of Metabolon's general practice, all columns were purchased from a single

manufacturer's lot at the outset of experiments. All solvents were similarly purchased in bulk from a single manufacturer's lot in sufficient quantity to complete all related experiments. For each sample, vacuum-dried samples were dissolved in injection solvent containing eight or more injection standards at fixed concentrations, depending on the platform. The internal standards were used both to assure injection and chromatographic consistency. Instruments were tuned and calibrated for mass resolution and mass accuracy daily.

The UPLC-MS/MS platform[38] utilized a Waters ACQUITY UPLC and a Thermo Scientific Q-Exactive high resolution/accurate mass spectrometer interfaced with a heated electrospray ionization (HESI-II) source and Orbitrap mass analyzer operated at 35,000 mass resolution. The sample extract was dried then reconstituted in solvents compatible to each method. Each reconstitution solvent contained a series of standards at fixed concentrations to ensure injection and chromatographic consistency. One aliquot was analyzed using acidic, positive ion-optimized conditions ("UPLC-MS/MS Pos"), and the other using basic, negative ion-optimized conditions ("UPLC-MS/MS Neg") in two independent injections using separate dedicated columns (Waters UPLC BEH C18-2.1 × 100 mm, 1.7 μm). Extracts reconstituted in acidic conditions were gradient-eluted using water and methanol containing 0.1% formic acid, while the basic extracts, which also used water/methanol, contained 6.5 mM ammonium bicarbonate. For the TB-ORD samples (i.e., TB vs. other respiratory diseases), one aliquot was analyzed using acidic positive ion conditions, chromatographically optimized for more hydrophilic compounds ("UPLC-MS/MS Pos Early"). In this method, the extract was gradient-eluted from a C18 column (Waters UPLC BEH C18-2.1 × 100 mm, 1.7 μm) using water and methanol, containing 0.05% perfluoropentanoic acid (PFPA) and 0.1% formic acid (FA). Another aliquot was also analyzed using acidic positive ion conditions; however, it was chromatographically optimized for more hydrophobic compounds ("UPLC-MS/MS Pos Late"). In this method, the extract was gradient-eluted from the same aforementioned C18 column using methanol, acetonitrile, water, 0.05% PFPA, and 0.01% FA and was operated at an overall higher organic content. Another aliquot was analyzed using basic negative ion optimized conditions using a separate dedicated C18 column ("UPLC-MS/MS Neg"). The basic extracts were gradient-eluted from the column using methanol and water, however with 6.5 mM ammonium bicarbonate at pH 8. The fourth aliquot was analyzed via negative ionization following elution from a HILIC column ("UPLC-MS/MS Polar" Waters UPLC BEH Amide 2.1 × 150 mm, 1.7 μm) using a gradient consisting of water and acetonitrile with 10 mM Ammonium Formate, pH 10.8. The MS analysis alternated between MS and data-dependent MSn scans using dynamic exclusion. The scan range varied slighted between methods but covered 80–1000 m/z.

For samples destined for analysis by GC-MS, an aliquot of extract was dried under vacuum desiccation for a minimum of 18 h prior to being derivatized under nitrogen using bistrimethyl-silyltrifluoroacetamide. Derivatized samples were separated on a 5% phenyldimethyl silicone column with helium as carrier gas and a temperature ramp from 60° to 340 °C within a 17-min period. All samples were analyzed on a Thermo-Finnigan Trace DSQ MS operated at unit mass resolving power with electron impact (EI) ionization and a 50–750 atomic mass unit scan range.

**Compound identification, quantification, and data curation**. Metabolites were identified by automated comparison of the ion features in the experimental samples to a reference library of chemical standard entries that included retention time, molecular weight (m/z), preferred adducts, and in-source fragments as well as associated MS spectra and curated by visual inspection for quality control using software developed at Metabolon[39]. Identification of known chemical entities is based on comparison with a spectral library of >4000 purified chemical standards. Commercially available purified standard compounds have been acquired and registered into LIMS for distribution to the UPLC-MS/MS and GC-MS platforms for determination of their detectable characteristics. Known metabolites reported in this study conform to the confidence Level 1 (the highest confidence level of identification) of the Metabolomics Standards Initiative[40,41], unless otherwise denoted with an asterisk. Additional mass spectral entries have been created for structurally unnamed biochemicals (> 5000 in the Metabolon library), which have been identified by virtue of their recurrent nature (both chromatographic and mass spectral). These compounds have the potential to be identified by future acquisition of a matching purified standard or by classical structural analysis.

Peaks were quantified using area-under-the-curve. Raw area counts for each metabolite in each sample were normalized to correct for variation resulting from instrument inter-day tuning differences by the median value for each run-day, therefore, setting the medians to 1.0 for each run. This preserved variation between samples but allowed metabolites of widely different raw peak areas to be compared on a similar graphical scale. Missing values were imputed with the observed minimum after normalization.

**Primary data**. Metabolic profiling was carried out for each site, using either serum or plasma samples. For a small number of samples, an insufficient amount of plasma was available, so the sample was diluted using RPMI buffer. The sample types taken from each study site are described in Supplementary Table 5. Metabolic profiling was carried out by Metabolon Inc. The metabolic profiles identified a total of 1701 unique metabolites. See Supplementary Data for details. Missing values

(not detected metabolites) were imputed using a Winsorization procedure (minimum value imputation).

The compatibility between serum and plasma samples was first tested using 36 time point/donor combinations for which both serum and plasma samples were available. The median correlation between the samples was 0.82 with 89% of the correlations significant (Pearson, $q < 0.05$) after correction for multiple testing. Furthermore, machine learning models trained on one sample type from the training set were tested on all other sample types in the training set (Supplementary Table 8).

**Data analysis planning**. In order to avoid data dredging and post-hoc bias, we first performed a detailed study on the training set only. Machine learning procedure was optimized and comparisons within cohort (Supplementary Table 6) and between cohorts (Supplementary Table 7) were conducted; differential abundance analysis with regression models was initially performed on the training set only. Based on the training set results, we put forward the hypothesis that we observe markers of progression toward TB rather than risk-associated factors, and accordingly planned the evaluation of the blinded set. Models were proposed and applied to the blinded test set. After unblinding, the performance of the prediction was evaluated for the regression models and repeated for the full dataset (as they were considered to have a primarily exploratory function).

**Machine learning**. We selected random forest machine learning because we have previously successfully applied these approaches to discriminate TB cases from healthy controls[19]. We used the random forest machine learning algorithm as implemented in the R package randomForest 4.6[42]. The cross-validation within a dataset was a modified k-fold procedure, in which all samples associated with an individual were removed in the cross-validation process. When models were trained and tested on different sample types (e.g., plasma samples were used to train a model and serum samples were used to test the model), the model was trained only on features which were found in both training and test sets, therefore the number of features may differ depending on the test set. The numbers of features used to train the models are shown in corresponding Supplementary Tables.

**Choosing a minimal classifer**. In order to select a minimum size classifier, various machine learning algorithms were applied using the interface available in the R package caret[43]. The algorithms chosen were random forests (caret – rf), generalized boosted models (caret – gbm), neural networks (caret – nnet), and elastic-net logistic regression (caret – glmnet). Each of these algorithms were trained on the entire training set, and performance evaluated using the LOOCV procedure described above. Model metabolites were ranked using the caret:varImp function, which uses model-specific methods to rank features. Random forest and boosted model importance was calculated as the increase in mean standard error on permutation of the metabolite, linear model importance was the magnitude of the metabolite coefficients, and neural network importance was calculated using Garson's algorithm to infer metabolite importance through network weights. After calculating variable importance for machine learning algorithms trained on all metabolites, models were retrained on the top 200 metabolites and importance recalculated. This was repeated recursively to create models with 100, 50, 25, and 10 metabolites. An optimal non site-specific small model was selected by only considering models with 50 or fewer metabolites, and calculating LOOCV AUCs for each site individually. The final model was selected by considering the poorest site-specific AUC from each model, i.e., the minimum site-specific AUC. The model with the highest minimum site-specific AUC was selected.

**Statistical analysis**. For each of the three sample types (plasma, serum, and plamsa diluted with RPMI), we removed metabolites with zero variance or without identification and selected only metabolites found in all three sample types. Then, we used a binomial regression model that included progressor/control grouping, stratification grouping, time to TB, and sample type, and considered the contrast of progressors vs. controls. We have used separate models to test the total dataset for metabolites common to all three sample types, and for serum, plasma and plasma/RPMI-specific metabolites. Furthermore, we have tested the dependence on time to diagnosis for progressor samples for all metabolites using a linear mixed model using the lme4 package[44].

For data standardization for visualization purposes, we fit a model that included all controlled variables except for the progressor/control grouping, and used the residuals from that model as standardized relative abundance score, and visualized the data with a LOESS fit.

To determine the compounds that were different between progressors and controls, we have used five linear models: (i) overall differences between progressors and controls, (ii) differences only for samples less than 5 months to diagnosis, (iii) three separate models for each of the three sample types (plasma, serum, and plasma/RPMI).

Enrichment analysis was performed using the tmod package[45] and metabolic profiling modules constructed from previously published metabolic profiles of TB patients and healthy donors[19].

**Code availability**. To ensure reproducibility of our findings, all data, scripts, and software packages necessary to replicate the results and generate the figures are available. The software versions used are available in Supplementary Table 18. The manuscript text itself has been automatically generated from an available computable document[46], explicitly stating all undertaken calculations starting with the primary data readouts. This manuscript has been prepared as an rmarkdown[47] document. The manuscript source code, including all analyses performed as well as data necessary to replicate all results and figures are available from the github platform (https://github.com/january3/gc6metabolomics).

## Data availability

Metabolomic data have been deposited to Metabolomic Workbench[48], ID PR000666 and are accessible under http://www.metabolomicsworkbench.org/data/DRCCMetadata.php?Mode=Project&ProjectID=PR000666. Data used to generate the figures 2–5 are available as a Source Data file.

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

## Acknowledgements

The Bill & Melinda Gates Foundation Grand Challenges in Global Health Program (BMGF GC6-74, #37772 and GC6-2013, OPP1055806), National Institutes of Health Contract No. HHSN266200700022C/NO1-AI-70022 for the Tuberculosis Research Unit (TBRU).

## Author contributions

Study concept and design: GC6-74 consortium (PI S.H.E.K., H.M.D., T.H.M.O., W.H.B., G.W., H.M.-K., R.H., W.A.H., S.K.P.). Sample collection: GC6-74 sites. Biobank, sample QC: S.S. Study design GC6-2013, metabolic profiling: S.H.E.K., J.M., J.W., J.S.S., T.J.S. GC6-2013 project management: G.W., S.H.E.K., T.J.S. Experimental design: J.W., D.E.Z., J.M. Data management and blinding: B.T., J.M., S.S. Metabolomics data collection: R.P. M. Data analysis: J.W., F.J.D., J.M., G.M., J.Z., E.T. ORD sample collection, management, biobank: J.S.S. Manuscript writing team: main writers: J.W., S.H.E.K., J.M.; further contributions: S.S., F.J.D., G.M., H.M.D., G.W., J.S.S., T.H.M.O., T.J.S., J.Z.

## Additional information

**Competing interests:** The authors declare no competing interests.

## The GC6-74 consortium

Almaz Abebe[14], Brian Abel[4], Richard Adegbola[2], Ifedayo Adetifa[2], Lyn Ambrose[15], Peter Andersen[16], Martin Antonio[2], Abraham Aseffa[8], Debbie van Baarle[17], Lew Barker[18], Yonas Bekele[8], Nicole Bilek[4], Gillian F. Black[11], Mark Bowmaker[4], Keith Branson[15], Michael Brennan[18], Novel N. Chegou[11], Femia Chilongo[15], William Kwong Chung[4], Tumani Corrah[2], Ameilia C. Crampin[15], Mark Doherty[16], Gregory Dolganov[19], Simon Donkor[2], Katrina Downing[4], Michelle Fisher[4], Kees L.M.C. Franken[10], Neil French[15], Larry Geiter[18], Annemieke Geluk[10], Robert Golinski[1], Patricia Gorak-Stolinska[9], Marielle C. Haks[10], Philip Hill[2], Jane Hughes[4], Yun-Gyoung Hur[9], Rachel Iwnetu[8], Marc Jacobson[1], Moses Joloba[7], Simone A. Joosten[10], Benjamin Kagina[4],

Desta Kassa[14], Hussein Kisingo[7], Michel R. Klein[10], Magdalena Kriel[11], Maeve Lalor[9], Ji-Sook Lee[9], Andre G. Loxton[11], Hassan Mahomed[4], Krista E. van Meijgaarden[10], Tsehayenesh Mesele[14], Frank Miedema[17], Adane Mihret[8], Humphrey Mulenga[4], Stefanie Muller[18], Hazzie Mvula[15], Nonhlanhla Nene[11], Bagrey Ngwira[15], Mary Nsereko[7], Brenda Okwera[7], Martin Ota[2], Adam Penn-Nicholson[4], Nelita Du Plessis[11], S. Ramachandran[18], Ida Rosenkrands[16], Jerry Sadoff[18], Jacky Saul[15], Gary Schoolnik[19], Felanji Simukonda[15], Donata Sizemore[18], Steven Smith[9], Anne Ben Smith[15], Gian van der Spuy[11], Kim Stanley[11], David Tabb[11], Mesfin Tafesse[8], Belete Tegbaru[14], Toyin Togun[2], Gerhardus Tromp[11], Tran Van[19], Kate Watkins[15], Frank Weichold[18], Karin Weldingh[16], Lawrence Yamuah[8] & Sarah Zalwango[7]

[14]Ethiopian Health & Nutrition Research Institute, P.O. box 1242 Addis Ababa, Ethiopia. [15]Karonga Prevention Study, P.O. Box 46, Chilumba, Malawi. [16]Department of Infectious Disease Immunology, Statens Serum Institut, DK-2300 Copenhagen S, Denmark. [17]University Medical Centre, 3508 GA Utrecht, The Netherlands. [18]Aeras, Rockville, MD 20850, USA. [19]Department of Microbiology and Immunology, Stanford University, Palo Alto, CA 94304, USA

