## [Peer Review File · Nature Communications]

Editorial Note: This manuscript has been previously reviewed at another journal that is not operating a transparent peer review scheme. This document only contains reviewer comments and rebuttal letters for versions considered at Nature Communications .

REVIEWERS' COMMENTS:

Reviewer #1 (Remarks to the Author):

This manuscript provides a valuable repository of clinically well curated metabolomic samples from a large multi-cohort natural history study of TB in subsaharan Africa. Using state-of-the-art analytical and statistical methods, followed by discriminatory reduction, the authors identify metabolic biosignatures potentially predictive of TB that could be detected prior to the onset of clinical sx suggestive of TB. They do so in 2 ways. The first is derived de novo (called risk-associated) and exhibits moderate performance characteristics that seem to vary according to study site/cohort. The second is based on a signature that had been previously identified in a comparison of patients with active TB vs controls (and hence a form of validation), which they call disease-associated. Both seem well controlled to be specific for TB and artifacts related to proximity to clinical disease.

However, the overall significance of these findings is unclear with respect to our understanding of TB biology or potential for clinical translation. In essence, these data provide proof-of-concept validation that specific metabolic signatures associated with TB and risk factors for TB exist. Unfortunately, the performance characteristics of the ones reported don't seem to be sufficiently robust to be of clear clinical value. In addition, they are not tied to defined concentrations and the magnitude of changes over time seem quite small. The significance of these signatures is further clouded by the fact that the risk-associated metabolites are reporters of biologically undefined processes that may or may not correspond to a combination of known risk factors (for example, could it be that they represent a combination of varying degrees of known risk factors). Given these ambiguities, it's unclear what to make of these data beyond the fact that there are metabolic biomarkers of TB risk and disease, in addition to already existing clinico-epidemiologic biomarkers. Absent such knowledge, it's unclear what the path forward is beyond simply retesting these same signatures (ie, how can we improve them).

On the other hand, the data represent a unique resource that may prove invaluable for future and thus require careful and extensive archiving in a way that maximizes public accessibility.

REVIEWERS' COMMENTS:

Reviewer #1 (Remarks to the Author):

This manuscript provides a valuable repository of clinically well curated metabolomic samples from a large multi-cohort natural history study of TB in subsaharan Africa. Using state-of-the-art analytical and statistical methods, followed by discriminatory reduction, the authors identify metabolic biosignatures potentially predictive of TB that could be detected prior to the onset of clinical sx suggestive of TB.

We thank the reviewer for appreciating our work.

They do so in 2 ways. The first is derived de novo (called risk-associated) and exhibits moderate performance characteristics that seem to vary according to study site/cohort. The second is based on a signature that had been previously identified in a comparison of patients with active TB vs controls (and hence a form of validation), which they call disease-associated. Both seem well controlled to be specific for TB and artifacts related to proximity to clinical disease.

However, the overall significance of these findings is unclear with respect to our understanding of TB biology or potential for clinical translation. In essence, these data provide proof-of-concept validation that specific metabolic signatures associated with TB and risk factors for TB exist. Unfortunately, the performance characteristics of the ones reported don't seem to be sufficiently robust to be of clear clinical value.

We agree with the reviewer that the study provides a first step towards a real-world application. However, we note that the signature fulfills the minimal requirements for a the target product profile for the development of a test for predicting progression to active TB disease as proposed by the WHO (2017), as stated in "Discussion" (p. 13, l. 321-324):

"In fact, for proximate samples, at 75% specificity, the sensitivity of these models approached (TB-HEALTHY, 73%) or exceeded (TB-ORD, 76%) the proposed requirements for a target product profile for the development of a test for predicting progression to active TB disease²¹."

In addition, they are not tied to defined concentrations and the magnitude of changes over time seem quite small. The significance of these signatures is further clouded by the fact that the risk-associated metabolites are reporters of biologically undefined processes that may or may not correspond to a combination of known risk factors (for example, could it be that they represent a combination of varying degrees of known risk factors).

We agree with the reviewer that the identified metabolic profiles need further work to be implemented, and that a path forward may be less straight-forward than is the case for transcriptomes, but potentially more rewarding. We have formulated our considerations as follows (p. 17, l. 410-414):

"Undoubtedly, before practical point-of-care application of a metabolomic signature further studies are needed. An important consideration for metabolomic studies is the availability

of inexpensive quantitative procedures for the metabolites of interest³⁶. As some of these are readily available (e.g. cortisol), a follow-up study should focus on those metabolites which can be determined by simple procedures, using our data set as a guide line."

Given these ambiguities, it's unclear what to make of these data beyond the fact that there are metabolic biomarkers of TB risk and disease, in addition to already existing clinico-epidemiologic biomarkers. Absent such knowledge, it's unclear what the path forward is beyond simply retesting these same signatures (ie, how can we improve them).

Transcriptomics aside, we are not aware of any other potentially implementable clinico-epidemiologic biomarkers which indicate subclinical TB 6 months before clinical diagnosis becomes possible.

On the other hand, the data represent a unique resource that may prove invaluable for future and thus require careful and extensive archiving in a way that maximizes public accessibility.

We thank the reviewer for recognizing the uniqueness of our work.